# TOWARDS BETTER EVALUATION OF GNN EXPRESSIVENESS WITH BREC DATASET

## ABSTRACT

Research on the theoretical expressiveness of Graph Neural Networks (GNNs) has developed rapidly, and many methods have been proposed to enhance the expressiveness. However, unifying all kinds of models into one framework is untractable, making it hard to measure and compare their expressiveness quantitatively. In contrast to theoretical analysis, another way to measure expressiveness is by evaluating model performance on certain datasets containing 1-WL-indistinguishable graphs. Previous datasets specifically designed for this purpose, however, face problems with difficulty (any model surpassing 1-WL has nearly 100% accuracy), granularity (models tend to be either 100% correct or near random guess), and scale (only several essentially different graphs in each dataset). To address these limitations, we propose a new expressiveness dataset, **BREC**, including 400 pairs of non-isomorphic graphs carefully selected from four primary categories (Basic, Regular, Extension, and CFI). These graphs have higher difficulty (up to 4-WL-indistinguishable), finer granularity (can compare models between 1-WL and 3-WL), and a larger scale (400 pairs or extend to 319600 pairs or even more). Further, we synthetically test 23 models with higher-than-1-WL expressiveness on our BREC dataset. Our experiment gives the first thorough measurement of the expressiveness of those state-of-the-art beyond-1-WL GNN models and reveals the gap between theoretical and practical expressiveness. We expect this dataset to serve as a benchmark for testing the expressiveness of future GNNs. Dataset and evaluation codes are released at: https://github.com/brec-iclr2024/brec-iclr2024.

## 1 INTRODUCTION

GNNs have been extensively utilized in bioinformatics, recommender systems, social networks, and others, yielding remarkable outcomes (Duvenaud et al., 2015; Barabási et al., 2011; Fan et al., 2019; Wang et al., 2018b; Berg et al., 2017; Zhou et al., 2020). Despite impressive empirical achievements, related investigations have revealed that GNNs exhibit limited abilities to distinguish some similar but non-isomorphic graphs. In practical situations, the inability to recognize specific structures, such as benzene rings (6-member rings), may cause misleading learning results. Xu et al. (2019); Morris et al. (2019) established a connection between the expressiveness of message-passing neural networks (MPNNs) and the WL test for graph isomorphism testing, demonstrating that MPNN's upper bound is 1-WL. Numerous subsequent studies have proposed GNN variants with enhanced expressiveness (Bevilacqua et al., 2022; Cotta et al., 2021; You et al., 2021; Zhang & Li, 2021).

Given the multitude of models employing different approaches, such as feature injection, equivariance maintenance, and subgraph extraction, a unified framework that can theoretically compare the expressive power among various variants is highly desirable. In this regard, several attempts have been made under the $k$-WL architecture. Maron et al. (2019b) propose the concept of $k$-order invariant/equivariant graph networks, which unify linear layers while preserving permutation invariance/equivariance. Additionally, Frasca et al. (2022) unify recent subgraph GNNs and establish that their expressiveness upper bound is 3-WL. Zhang et al. (2023a) further construct a comprehensive expressiveness hierarchy for subgraph GNNs. Nonetheless, the magnitude of the gaps remains unknown. Furthermore, there exist methods that are difficult to categorize within the $k$-WL hierarchy. For instance, Papp & Wattenhofer (2022) propose four extensions of GNNs, each of which cannot strictly compare with the other. Similarly, Feng et al. (2022) propose a GNN that is partially stronger than 3-WL yet fails to distinguish many 3-WL-distinguishable graphs. In a different ap-

proach, Huang et al. (2023) propose evaluating expressiveness by enumerating specific significant substructures, such as 6-cycles. Zhang et al. (2023b) introduces biconnectivity as a measurement.

Without a unified theoretical characterization of expressiveness, employing expressiveness datasets for testing proves valuable. Notably, three expressiveness datasets, EXP, CSL, and SR25, have been introduced by Abboud et al. (2021); Murphy et al. (2019); Balcilar et al. (2021) and have found widespread usage in recent studies. However, these datasets exhibit notable limitations. Firstly, they lack sufficient difficulty. The EXP and CSL datasets solely consist of examples where 1-WL fails, and most recent GNN variants have achieved perfect accuracy on these datasets. Secondly, the granularity of these datasets is too coarse, which means that graphs in these datasets are generated using a single method, resulting in a uniform level of discrimination difficulty. Consequently, the performance of GNN variants often falls either at random guessing (completely indistinguishable) or 100% (completely distinguishable), thereby hindering the provision of a nuanced measure of expressiveness. Lastly, these datasets suffer from small sizes, typically comprising only a few substantially different graphs, raising concerns of incomplete measurement.

To overcome the limitations of previous datasets, we propose BREC, including 400 pairs of non-isomorphic graphs in 4 major categories: Basic, Regular, Extension, and CFI graphs. Compared to previous datasets, BREC has a greater difficulty (up to 4-WL-indistinguishable), finer granularity (can compare models between 1-WL and 3-WL), and larger scale (800 non-isomorphic graphs organized as 400 pairs or extend to 319600 pairs or even more), addressing previous ones' shortcomings.

Due to the increased size and diversity of the dataset, the traditional classification task may not be suitable for training-based evaluation methods that rely on generalization ability. Thus, we propose a novel evaluation procedure based on directly comparing the discrepancies between model outputs to test pure practical expressiveness. Acknowledging the impact of numerical precision owing to tiny differences between graph pairs, we propose reliable paired comparisons building upon a statistical method (Fisher, 1992; Johnson & Wichern, 2007), which offers a precise error bound. Experiments verify that the evaluation procedure aligns well with known theoretical results.

Finally, we comprehensively compared 23 representative beyond-1-WL models on BREC. Our experiments first give a **reliable empirical comparison** of state-of-the-art GNNs' expressiveness. The currently most thorough investigation is a good start for gaining deeper insights into various schemes to enhance GNNs' expressiveness. On BREC, GNN accuracies range from 41.5% to 70.2%, with $I^2$-GNN (Huang et al., 2023) performing the best. The 70.2% highest accuracy also implies that the dataset is **far from saturation**. We expect BREC can serve as a benchmark for testing future GNNs' expressiveness. Our dataset is included in https://github.com/brec-iclr2024/brec-iclr2024.

## 2 LIMITATIONS OF EXISTING DATASETS

**Preliminary.** We utilize the notation $\{\}$ to represent sets and $\{\{\}\}$ to represent multisets. The cardinality of a (multi)set $\mathbb{S}$ is denoted as $|\mathbb{S}|$. The index set is denoted as $[n] = 1, \ldots, n$. A graph is denoted as $\mathcal{G} = (\mathbb{V}(\mathcal{G}), \mathbb{E}(\mathcal{G}))$, where $\mathbb{V}(\mathcal{G})$ represents the set of *nodes* or *vertices* and $\mathbb{E}(\mathcal{G})$ represents the set of *edges*. Without loss of generality, we assume $|\mathbb{V}(\mathcal{G})| = n$ and $\mathbb{V}(\mathcal{G}) = [n]$.

The permutation or reindexing of $\mathcal{G}$ is denoted as $\mathcal{G}^\pi = (\mathbb{V}(\mathcal{G}^\pi), \mathbb{E}(\mathcal{G}^\pi))$ with the permutation function $\pi : [n] \to [n]$, s.t. $(u, v) \in \mathbb{E}(\mathcal{G}) \iff (\pi(u), \pi(v)) \in \mathbb{E}(\mathcal{G}^\pi)$. Here node and edge features are excluded from definitions for briefness. Additional discussions can be found in Appendix B.

**Graph Isomorphism (GI) Problem.** Two graphs $\mathcal{G}$ and $\mathcal{H}$ are considered isomorphic (denoted as $\mathcal{G} \simeq \mathcal{H}$) if $\exists\, \phi$(a bijection mapping) $: \mathbb{V}(\mathcal{G}) \to \mathbb{V}(\mathcal{H})$ s.t. $(u, v) \in \mathbb{E}(\mathcal{G})$ iff. $(\phi(u), \phi(v)) \in \mathbb{E}(\mathcal{H})$. GI is essential in expressiveness. Only if GNN successfully distinguishes two non-isomorphic graphs can they be assigned different labels. Some researchers (Chen et al., 2019; Geerts & Reutter, 2022) indicate the equivalence between GI and function approximation, underscoring the importance of GI. However, we currently do not have polynomial-time algorithms for solving the GI problem. A naive solution involves iterating all $n!$ permutations to test whether such a bijection exists.

**Weisfeiler-Lehman algorithm (WL).** WL is a well-known isomorphism test relying on color refinement (Weisfeiler & Leman, 1968). In each iteration, WL assigns a state (or color) to each node by aggregating information from its neighboring nodes' states. This process continues until convergence, resulting in a multiset of node states representing the final graph representation. While WL

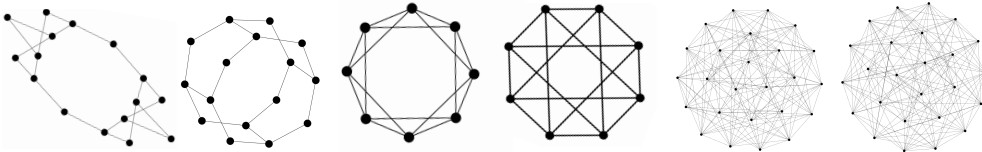

(a) EXP dataset core pair sample   (b) CSL graph($m = 10, r = 2/3$)   (c) SR25 dataset sample

Figure 1: Sample graphs in previous datasets

Table 1: Dataset statistics

| Dataset | # Graphs | # Core graphs[a] | # Nodes | Distinguishing difficulty | Evaluation metrics |
|---|---|---|---|---|---|
| EXP | 1200 | 6 | 33-73 | 1-WL-indistinguishable | 2-way classification |
| CSL | 150 | 10 | 41 | 1-WL-indistinguishable | 10-way classification |
| SR25 | 15 | 15 | 25 | 3-WL-indistinguishable | 15-way classification |
| **BREC** | **800** | **800** | **10-198** | **1-WL to 4-WL-indistinguishable** | **Reliable Paired Comparisons** |

[a] Core graphs represent graphs that actually serve to measure expressiveness.

effectively identifies most non-isomorphic graphs, it may fail in certain simple graphs, leading to the development of extended versions. One such extension is $k$-WL, which treats each $k$-tuple of nodes as a unit for aggregating information. Another slightly different method (Cai et al., 1989) is also referred to as $k$-WL. To avoid confusion, we follow Morris et al. (2019) to call the former $k$-WL and the latter $k$-FWL. Further information can be found in Appendix C.

Given the significance of GI and WL, several expressiveness datasets have been introduced, with the following three being the most frequently utilized. We selected a pair of graphs from each dataset, illustrated in Figure 1. Detailed statistics for these datasets are presented in Table 1.

**EXP Dataset.** This dataset is generated pairwise. Each graph in a pair includes two disconnected components, the core component and planar component, where the former are two 1-WL-indistinguishable counterexamples while the latter are identical and only for adding noise. The two graphs are labeled 0/1 based on whether their core component satisfies the SAT condition for a binary classification problem. Although it is formally consistent with general datasets, the insufficient difficulty and number of different core components (**only three substantially different pairs**) results in most recent GNNs achieving nearly 100% accuracy, making detailed comparisons unavailable.

**CSL Dataset.** This dataset consists of Circulant Skip Links (CSL) graphs, which are 1-WL-indistinguishable 4-degree regular graphs. Ten distinct CSL graphs with 41 nodes are generated first. Each of the ten distinct CSL graphs is treated as a separate class, and the task is to train a 10-way classification model. Then, each graph is reindexed 14 times, resulting in the final dataset with 150 graphs. Because of the relatively low difficulty and their fixed structure (only **ten essentially different graphs with the same number of nodes and degree**), many recent expressive GNN models achieve close to 100% accuracy. More details about CSL graphs are in Appendix D.

**SR25 Dataset.** This dataset consists of 15 3-WL-indistinguishable Strongly Regular Graphs (SR) with parameter srg(25,12,5,6)[1]. In practice, SR25 is transformed into a 15-way classification problem for mapping each graph into a different class where the training and test graphs overlap. Most methods obtain 6.67% (1/15) accuracy due to 3-WL's high expressiveness. However, some methods partially surpassing 3-WL achieve 100% accuracy easily since each graph has the same parameters.

These three datasets have limitations regarding difficulty, granularity, and scale. In terms of difficulty, they are all bounded by 3-WL, failing to evaluate models (partly) beyond 3-WL (Feng et al., 2022). In terms of granularity, the graphs are generated in one way with repetive parameters, which easily leads to a 0/1 step function of model performance and cannot measure subtle differences between models. In terms of scale, the number of substantially different graphs in the datasets is small, and the test results may be incomplete to reflect expressiveness measurement.

## 3 BREC: A NEW DATASET FOR EXPRESSIVENESS

We propose a new expressiveness dataset, BREC, to address the limitations regarding difficulty, granularity, and scale. It consists of four major categories of graphs: Basic, Regular, Extension, and CFI. Basic graphs include relatively simple 1-WL-indistinguishable graphs. Regular graphs include

---

[1]Strongly regular graphs can be described by four parameters. More details can be found in Appendix A

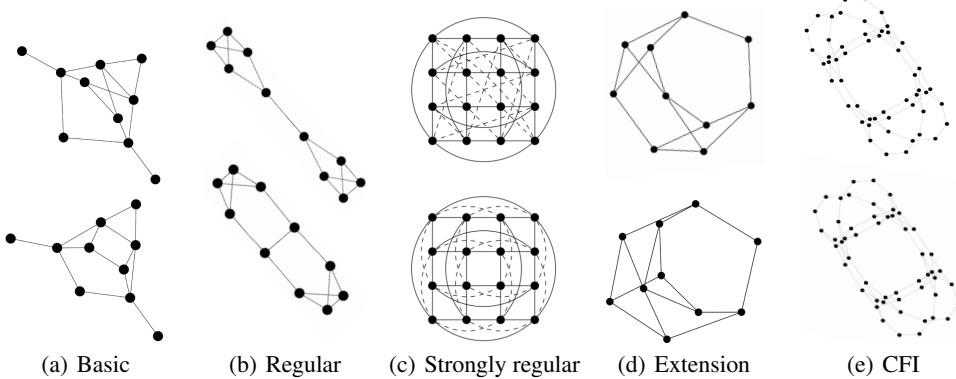

| (a) Basic | (b) Regular | (c) Strongly regular | (d) Extension | (e) CFI |

Figure 2: BREC dataset samples

four types of subcategorized regular graphs. Extension graphs include special graphs that arise when comparing four kinds of GNN extensions (Papp & Wattenhofer, 2022). CFI graphs include graphs generated by CFI methods[2] (Cai et al., 1989) with high difficulty. Some samples are shown in Fig 2.

## 3.1 DATASET COMPOSITION

BREC includes 800 non-isomorphic graphs arranged in a pairwise manner to construct 400 pairs, with detailed composition as follows: (For detailed generation process, please refer to Appendix L)

**Basic Graphs.** Basic graphs consist of 60 pairs of 10-node graphs. These graphs are collected from an exhaustive search and intentionally designed to be non-regular. Although they are 1-WL-indistinguishable, most can be distinguished by expressive GNN variants. Basic graphs can also be regarded as an augmentation of the EXP dataset, as they both employ non-regular 1-WL-indistinguishable graphs. Nevertheless, Basic graphs offer a greater abundance of instances and more intricate graph patterns. The relatively small size also facilitates visualization and analysis.

**Regular Graphs.** Regular graphs consist of 140 pairs of regular graphs, including 50 pairs of simple regular graphs, 50 pairs of strongly regular graphs, 20 pairs of 4-vertex condition graphs, and 20 pairs of distance regular graphs. A regular graph refers to a graph where all nodes possess the same degree. Regular graphs are 1-WL-indistinguishable, and some studies delve into the analysis of GNN expressiveness from this perspective (Li et al., 2020; Zhang & Li, 2021). We denote regular graphs without any special properties as simple regular graphs. When exploring more intricate regular graphs, the concept of strongly regular graphs (where 3-WL fails) is often introduced. They further require that the number of neighboring nodes shared by any two nodes depends solely on their connectivity. Notable examples of strongly regular graphs include the $4 \times 4$-Rook's graph and the Shrikhande graph (Fig 2(c)). Additionally, the $4 \times 4$-Rook's graph satisfies the 4-vertex condition property, which signifies that the number of connected edges between the common neighbors of any two nodes is solely determined by their connectivity (Brouwer et al., 2023). It is worth mentioning that the diameter of a connected strongly regular graph is always 2 (Brouwer et al., 2012b). A more challenging type of graph known as the distance regular graphs (Brouwer et al., 2012a) is proposed aiming for extending the diameter. By expanding upon the existing subdivisions of regular graphs, this section widens the range of difficulty and complexity. Moreover, unlike the previous datasets, regular graphs are not limited to sharing identical parameters for all graphs within each category, greatly enhancing diversity. More details about regular graphs can be found in Appendix A.

**Extension Graphs.** Extension graphs consist of 100 pairs of graphs inspired by Papp & Wattenhofer (2022). They proposed 4 types of theoretical GNN extensions: $k$-WL hierarchy-based, substructure-counting-based, $k$-hop-subgraph-based, and marking-based without strict comparison relationship. Leveraging the insights from theoretical analysis and some empirically derived findings, we generated 100 pairs of graphs between 1-WL and 3-WL distinguishing difficulty to improve granularity. Note that we are the first to realize the algorithms and generate counterexamples on a large scale.

**CFI Graphs.** CFI graphs consist of 100 pairs of graphs inspired by Cai et al. (1989). They developed a method to generate graphs distinguishable by $k$-WL but not by $(k-1)$-WL for any $k$. We are the

---

[2]CFI is short for Cai-Furer-Immerman algorithm, which can generate counterexample graphs for any k-WL.

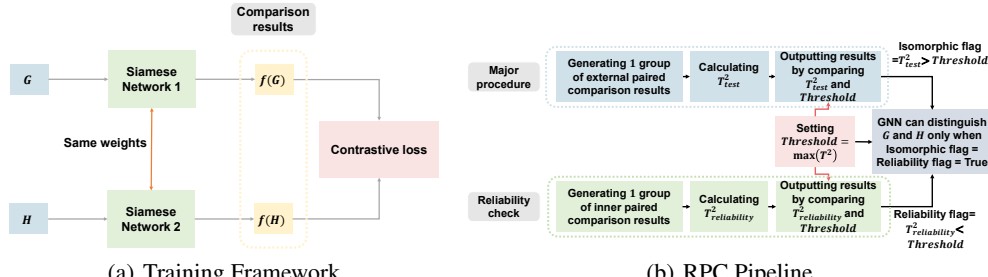

Figure 3: Evaluation Method

first to implement it and create 100 pairs of graphs spanning up to 4-WL-indistinguishable, even surpassing the current research's upper bounds. Specifically, 60 pairs are solely distinguishable by 3-WL, 20 are solely distinguishable by 4-WL, and 20 are even 4-WL-indistinguishable. As the most challenging part, it pushes the upper limit of difficulty even higher. Furthermore, the graph sizes in this section are larger than other parts (up to 198 nodes). This aspect intensifies the challenge of the dataset, demanding a model's ability to process graphs with heterogeneous sizes effectively.

## 3.2 ADVANTAGES

**Difficulty.** The CFI graphs raise difficulty to 4-WL-indistinguishable. The newly involved 4-vertex condition and distance regular graphs also pose greater challenges regarding higher regularity.

**Granularity.** The different classes of graphs in BREC exhibit varying difficulty levels, each contributing to the dataset in distinct ways. Basic graphs contain fundamental 1-WL-indistinguishable graphs, similar to the EXP dataset, as a starting point for comparison. Regular graphs extend the CSL and SR25 datasets. The major components of regular graphs are simple regular graphs and strongly regular graphs, where 1-WL and 3-WL fail, respectively. Including 4-vertex condition graphs and distance regular graphs further elevates the complexity. Extension graphs bridge the gap between 1-WL and 3-WL, offering a finer-grained comparison for evaluating models beyond 1-WL. CFI graphs span the spectrum of difficulty from 1-WL to 4-WL-indistinguishable. By comprehensive graph composition, BREC explores the boundaries of graph pattern distinguishability.

**Scale.** While previous datasets relied on only a few essentially different graphs, BREC utilizes a collection of 800 different graphs organized as 400 pairs. This significant increase in the number of graphs greatly enhances the diversity. The larger graph set in BREC also contributes to a more varied distribution of graph statistics. In contrast, previous datasets such as CSL and SR25 only have the same number of nodes and degrees across all graphs. For detailed statistics of BREC, please refer to Appendix E. BREC can also be easily further scaled up to 319600 pairs by iterating over all possible compare combinations, or even more by adding more graphs as it is a small sample of what we have done in the generation process (Appendix L). However, we deliberately did not scale it to facilitate a lower testing burden and balance the distribution of graphs with different difficulties. The experiments also verify that the current size is enough to find subtle differences between models.

## 4  RPC: A NEW EVALUATION METHOD

This section introduces a novel training framework and evaluation method for BREC. Unlike previous datasets, BREC departs from the conventional classification setting, where each graph is assigned a label, a classification model is trained, and the accuracy on test graphs serves as the measure of expressiveness. The labeling schemes used in previous datasets like semantic labels based on SAT conditions in EXP, or distinct labels for essentially different graphs in CSL and SR25, do not apply to BREC. There are two primary reasons. First, BREC aims to enrich the diversity of graphs, which precludes using a semantic label tied to SAT conditions, as it would significantly limit the range of possible graphs. Second, assigning a distinct label to each graph in BREC would result in an 800-class classification problem, where performance could be influenced by factors other than expressiveness. Our core idea is to measure models' practical "separating power" directly. Thus BREC is organized in pairs, where each pair is individually tested to determine whether a GNN can distinguish them. By adopting a pairwise evaluation method, BREC provides a more focused measure of models' expressiveness, aligning to assess distinguishing ability.

Nevertheless, how can we say a pair of graphs is successfully distinguished? Previous researchers tend to set a small threshold (like 1E-4) manually. If the embedding distance between them is larger than the threshold, the GNN is considered can distinguish them. This setting may satisfy previous datasets' requirements due to the relatively simple construction. However, it lacks **reliability** on numerical precision with more complex graphs where tiny differences may even overlap with numerical fluctuations. In order to yield dependable outcomes, we propose an evaluation method measuring both **external difference** and **internal fluctuations**. Furthermore, we introduce a training framework for pairwise data, employing the siamese network design (Koch et al., 2015) and contrastive loss (Hadsell et al., 2006; Wang et al., 2018a). The pipeline is depicted in Fig 3(a).

## 4.1 TRAINING FRAMEWORK

We adhere to the siamese network design (Koch et al., 2015) to train a model to distinguish each pair of graphs. The central component consists of two identical models maintaining identical parameters. For a pair of graphs inputted, it outputs a pair of embeddings. Subsequently, the difference between them is assessed using cosine similarity. The loss function is formulated as follows:

$$L(f, \mathcal{G}, \mathcal{H}) = \text{Max}(0, \frac{f(\mathcal{G}) \cdot f(\mathcal{H})}{||f(\mathcal{G})|| \, ||f(\mathcal{H})||} - \gamma),$$ (1)

where the GNN model $f : \{\mathcal{G}\} \to \mathbb{R}^d$, $\mathcal{G}$ and $\mathcal{H}$ are two graphs, and $\gamma$ is a margin hyperparameter (set to 0 in our experiments). The loss function aims to promote the cosine similarity value lower than $\gamma$, thereby encouraging a greater separation between the two graph embeddings.

The training process **yields several benefits** for the models. Firstly, it helps the GNN to achieve its theoretical expressiveness. The GNN expressiveness analysis focuses primarily on the network's structure without imposing any constraints on its parameters, which means it is exploring the expressiveness of **a group of functions**. If a model with particular parameters can distinguish a pair of graphs, the model's design and structure are considered possessing sufficient expressiveness. However, it is impractical to iterate all possible parameter combinations to test the real upper bound. Hence, training can **realize searching** in the function space, enabling models to achieve better practical expressiveness. Furthermore, training aids components to **possess specific properties**, such as injectivity and universal approximation, which are vital for attaining theoretical expressiveness. These properties require specific parameter configurations, but randomly initialized parameters may not satisfy. Moreover, through training, model-distinguishable pairs are **more easily discriminated** from model-indistinguishable pairs, which helps reducing the false negative rate caused by numerical precision. The difference between model-distinguishable pairs' embeddings is further magnified in the pairwise contrastive training process. However, the difference for model-indistinguishable pairs caused by numerical precision remains unaffected mainly. The framework is shown in Fig 3(a).

## 4.2 EVALUATION METHOD

We evaluate models by comparing the outputs of two non-isomorphic graphs. If we notice a significant difference between the outputs, we conclude that the GNN can distinguish the pair of graphs. However, setting a suitable threshold can be challenging. A large threshold may yield false negatives, which means the model can distinguish the pair, but the observed difference falls short of the threshold. Conversely, a small threshold may yield false positives, which means the model cannot distinguish the pair, but fluctuating errors cause the difference to exceed the threshold.

To address the issue of fluctuating errors, we draw inspiration from Paired Comparisons (Fisher, 1992). It involves comparing two groups of results instead of a single pair. The influence of random errors is mitigated by repeatedly generating results and comparing the two groups of results. Building upon it, we introduce a method called **R**eliable **P**aired **C**omparison (RPC) to verify whether a GNN genuinely produces distinct outputs for a pair of graphs. The pipeline is depicted in Fig 3(b).

RPC consists of two main components: Major procedure and Reliability check. The Major procedure is conducted on a pair of non-isomorphic graphs to measure their dissimilarity. In contrast, the Reliability check is conducted on graph automorphisms to capture internal fluctuations.

**Major procedure.** Given two non-isomorphic graphs $\mathcal{G}, \mathcal{H}$, we create $q$ copies of each by random permutation (still isomorphic to original graph) to generate two groups of graphs, denoted as:

$$\mathcal{G}_i, \, \mathcal{H}_i, \, i \in [q].$$ (2)

Supposing the GNN $f : \{\mathcal{G}\} \to \mathbb{R}^d$, we first calculate $q$ differences utilizing Paired Comparisons.

$$\boldsymbol{d}_i = f(\mathcal{G}_i) - f(\mathcal{H}_i),\ i \in [q]. \tag{3}$$

**Assumption 4.1** $\boldsymbol{d}_i$ *are independent* $\mathcal{N}(\boldsymbol{\mu}, \boldsymbol{\Sigma})$ *random vectors.*

The above assumption is based on a more basic assumption that $f(\mathcal{G}_i),\ f(\mathcal{H}_i)$ follow Gaussian distributions, which presumes that random permutation only introduces Gaussian noise to the result.

If the GNN cannot distinguish $\mathcal{G}$ and $\mathcal{H}$, the mean difference should satisfy $\boldsymbol{\mu} = \boldsymbol{0}$. To check whether the equation holds, we can conduct an $\alpha$-level Hotelling's T-square test, comparing the hypotheses $H_0 : \boldsymbol{\mu} = \boldsymbol{0}$ against $H_1 : \boldsymbol{\mu} \neq \boldsymbol{0}$. The $T^2$-statistic for $\boldsymbol{\mu}$ is calculated as follows:

$$T^2 = q(\overline{\boldsymbol{d}} - \boldsymbol{\mu})^T \boldsymbol{S}^{-1}(\overline{\boldsymbol{d}} - \boldsymbol{\mu}), \tag{4}$$

where

$$\overline{\boldsymbol{d}} = \frac{1}{q}\sum_{i=1}^{q}\boldsymbol{d}_i,\ \boldsymbol{S} = \frac{1}{q-1}\sum_{i=1}^{q}(\boldsymbol{d}_i - \overline{\boldsymbol{d}})(\boldsymbol{d}_i - \overline{\boldsymbol{d}})^T. \tag{5}$$

Hotelling's T-square test proves that $T^2$ is distributed as an $\frac{(q-1)d}{q-d}F_{d,q-d}$ random variable, where $F_{d,q-d}$ represents $F$-distribution with degree of freedom $d, q - d$ (Hotelling, 1992). The theorem establishes a connection between the unknown parameter $\boldsymbol{\mu}$ and a definite distribution $F_{d,q-d}$, allowing us to confirm the confidence interval of $\boldsymbol{\mu}$ by testing the distribution fit. To test the hypothesis $H_0 : \boldsymbol{\mu} = \boldsymbol{0}$, we substitute $\boldsymbol{\mu} = \boldsymbol{0}$ into Equation (4), obtaining $T^2_{\text{test}} = q\overline{\boldsymbol{d}}^T \boldsymbol{S}^{-1}\overline{\boldsymbol{d}}$. Then an $\alpha$-level test of $H_0 : \boldsymbol{\mu} = \boldsymbol{0}$ versus $H_1 : \boldsymbol{\mu} \neq \boldsymbol{0}$ accepts $H_0$ (the GNN cannot distinguish the pair) if:

$$T^2_{\text{test}} = q\overline{\boldsymbol{d}}^T \boldsymbol{S}^{-1}\overline{\boldsymbol{d}} < \frac{(q-1)d}{(q-d)}F_{d,q-d}(\alpha), \tag{6}$$

where $F_{d,q-d}(\alpha)$ is the upper $(100\alpha)$th percentile of the $F$-distribution $F_{d,q-d}$ (Fisher, 1950) with $d$ and $q - d$ degrees of freedom. Similarly, we reject $H_0$ (the GNN can distinguish the pair) if

$$T^2_{\text{test}} = q\overline{\boldsymbol{d}}^T \boldsymbol{S}^{-1}\overline{\boldsymbol{d}} > \frac{(q-1)d}{(q-d)}F_{d,q-d}(\alpha). \tag{7}$$

**Reliability check.** With an appropriate choice of $\alpha$, the Major procedure provides a dependable confidence interval for assessing the distinguishability. However, manually selected $\alpha$ based on heuristics may not be optimal. Furthermore, computational precisions can introduce distribution shifts in assumed Gaussian fluctuations. To address this issue, we introduce the Reliability check. It bridges external differences between two graphs and internal fluctuations within a single graph.

WLOG, we replace $\mathcal{H}$ by permutation of $\mathcal{G}$, i.e., $\mathcal{G}^{\pi}$. We can then obtain the internal fluctuations within $\mathcal{G}$ by comparing it with $\mathcal{G}^{\pi}$, and the external difference between $\mathcal{G}$ and $\mathcal{H}$ by comparing $\mathcal{G}$ and $\mathcal{H}$. We utilize the same step as Major procedure on $\mathcal{G}$ and $\mathcal{G}^{\pi}$, calculating the $T^2$-statistics as:

$$T^2_{\text{reliability}} = q\overline{\boldsymbol{d}}^T \boldsymbol{S}^{-1}\overline{\boldsymbol{d}}, \tag{8}$$

where $\overline{\boldsymbol{d}} = \frac{1}{q}\sum_{i=1}^{q}\boldsymbol{d}_i,\ \boldsymbol{d}_i = f(\mathcal{G}_i) - f(\mathcal{G}_i^{\pi}),\ i \in [q],\ \boldsymbol{S} = \frac{1}{q-1}\sum_{i=1}^{q}(\boldsymbol{d}_i - \overline{\boldsymbol{d}})(\boldsymbol{d}_i - \overline{\boldsymbol{d}})^T. \tag{9}$

Recalling that $\mathcal{G}$ and $\mathcal{G}^{\pi}$ are isomorphic, the GNN should not distinguish between them, implying that $\boldsymbol{\mu} = \boldsymbol{0}$. Therefore, the test is considered reliable only if $T^2_{\text{reliability}} < \frac{(q-1)d}{(q-d)}F_{d,q-d}(\alpha)$. Combining the reliability and distinguishability results, we get the complete RPC (Fig 3(b)) as follows:

For each pair of graphs $\mathcal{G}$ and $\mathcal{H}$, we first calculate the threshold value, denoted as Threshold $= \frac{(q-1)d}{(q-d)}F_{d,q-d}(\alpha)$. Next, we conduct the Major procedure on $\mathcal{G}$ and $\mathcal{H}$ for distinguishability and perform the Reliability check on $\mathcal{G}$ and $\mathcal{G}^{\pi}$ for Reliability. Only when the $T^2$-statistic from the Major procedure, denoted as $T^2_{\text{test}}$, and the $T^2$-statistic from the Reliability check, denoted as $T^2_{\text{reliability}}$, satisfying $T^2_{\text{reliability}} < \text{Threshold} < T^2_{\text{test}}$, do we conclude that the GNN can distinguishing $\mathcal{G}$ and $\mathcal{H}$.

We further propose **R**eliable **A**daptive **P**aired **C**omparisons (RAPC), aiming to adaptively adjust the threshold and provide an upper bound for false positive rates. In practice, we use **RPC** due to its less computational time and satisfactory performance. For more details, please refer to Appendix F.

Table 2: Pair distinguishing accuracies on BREC

| Type | Model | Basic Graphs (60) | | Regular Graphs (140) | | Extension Graphs (100) | | CFI Graphs (100) | | Total (400) | |
|---|---|---|---|---|---|---|---|---|---|---|---|
| | | Number | Accuracy | Number | Accuracy | Number | Accuracy | Number | Accuracy | Number | Accuracy |
| Non-GNNs | 3-WL | 60 | 100% | 50 | 35.7% | 100 | 100% | 60 | 60.0% | 270 | 67.5% |
| | SPD-WL | 16 | 26.7% | 14 | 11.7% | 41 | 41% | 12 | 12% | 83 | 20.8% |
| | $S_3$ | 52 | 86.7% | 48 | 34.3% | 5 | 5% | 0 | 0% | 105 | 26.2% |
| | $S_4$ | 60 | 100% | 99 | 70.7% | 84 | 84% | 0 | 0% | 243 | 60.8% |
| | $N_1$ | 60 | 100% | 99 | 85% | 93 | 93% | 0 | 0% | 252 | 63% |
| | $N_2$ | 60 | 100% | 138 | 98.6% | 100 | 100% | 0 | 0% | 298 | 74.5% |
| | $M_1$ | 60 | 100% | 50 | 35.7% | 100 | 100% | 41 | 41% | 251 | 62.8% |
| Subgraph GNNs | NGNN | 59 | 98.3% | 48 | 34.3% | 59 | 59% | 0 | 0% | 166 | 41.5% |
| | DE+NGNN | 60 | 100% | 50 | 35.7% | 100 | 100% | 21 | 21% | 231 | 57.8% |
| | DS-GNN | 58 | 96.7% | 48 | 34.3% | 100 | 100% | 16 | 16% | 222 | 55.5% |
| | DSS-GNN | 58 | 96.7% | 48 | 34.3% | 100 | 100% | 15 | 15% | 221 | 55.2% |
| | SUN | 60 | 100% | 50 | 35.7% | 100 | 100% | 13 | 13% | 223 | 55.8% |
| | SSWL_P | 60 | 100% | 50 | 35.7% | 100 | 100% | 38 | 38% | 248 | 62% |
| | GNN-AK | 60 | 100% | 50 | 35.7% | 97 | 97% | 15 | 15% | 222 | 55.5% |
| | KP-GNN | 60 | 100% | 106 | 75.7% | 98 | 98% | 11 | 11% | 275 | 68.8% |
| | I$^2$-GNN | 60 | 100% | 100 | 71.4% | 100 | 100% | 21 | 21% | 281 | 70.2% |
| k-WL GNNs | PPGN | 60 | 100% | 50 | 35.7% | 100 | 100% | 23 | 23% | 233 | 58.2% |
| | $\delta$-k-LGNN | 60 | 100% | 50 | 35.7% | 100 | 100% | 6 | 6% | 216 | 54% |
| | KC-SetGNN | 60 | 100% | 50 | 35.7% | 100 | 100% | 1 | 1% | 211 | 52.8% |
| Substructure GNNs | GSN | 60 | 100% | 99 | 70.7% | 95 | 95% | 0 | 0% | 254 | 63.5% |
| Random GNNs | DropGNN | 52 | 86.7% | 41 | 29.3% | 82 | 82% | 2 | 2% | 177 | 44.2% |
| | OSAN | 56 | 93.3% | 8 | 5.7% | 79 | 79% | 5 | 5% | 148 | 37% |
| Transformer GNNs | Graphormer | 16 | 26.7% | 12 | 8.6% | 41 | 41% | 10 | 10% | 79 | 19.8% |

## 5 EXPERIMENT

In this section, we evaluate the expressiveness of 23 representative models using our BREC dataset.

**Model selection.** We evaluate six categories of methods: non-GNN methods, subgraph-based GNNs, $k$-WL-hierarchy-based GNNs, substructure-based GNNs, transformer-based GNNs, and random GNNs. Our primary focus will be on the first three categories. We implement four types of non-GNN baselines based on Papp & Wattenhofer (2022); Ying et al. (2021), including WL test (3-WL and SPD-WL), counting substructures ($S_3$ and $S_4$), neighborhood up to a certain radius ($N_1$ and $N_2$), and marking ($M_1$). We implemented them by adding additional features during the WL test update or using heterogeneous message passing. It is important to note that they are more theoretically significant than practical since they may require exhaustive enumeration or exact isomorphism encoding of various substructures. We additionally included 16 state-of-the-art GNNs, including NGNN (Zhang & Li, 2021), DE+NGNN (Li et al., 2020), DS/DSS-GNN (Bevilacqua et al., 2022), SUN (Frasca et al., 2022), SSWL_P (Zhang et al., 2023a), GNN-AK (Zhao et al., 2022a), KP-GNN (Feng et al., 2022), I$^2$-GNN (Huang et al., 2023), PPGN (Maron et al., 2019a), $\delta$-k-LGNN (Morris et al., 2020), KC-SetGNN (Zhao et al., 2022b), GSN (Bouritsas et al., 2022), DropGNN (Papp et al., 2021), OSAN (Qian et al., 2022), and Graphormer (Ying et al., 2021).

Table 2 presents the primary results. $N_2$ achieves the highest accuracy among non-GNNs, and I$^2$-GNN achieves the highest among GNNs. We provide a detailed analysis of each method's accuracy across various graphs. The findings generally indicate that practical expressiveness aligns closely with theoretical expectations. Nevertheless, there are still some situations where a gap between theory and practice persists. Detailed experiment settings are included in Appendix K.

**Non-GNN baselines.** 3-WL successfully distinguishes all Basic graphs, Extension graphs, simple regular graphs and 60 CFI graphs as expected. $S_3$, $S_4$, $N_1$, and $N_2$ demonstrate excellent performance on small-radius graphs such as Basic, Regular, and Extension graphs. However, due to their limited receptive fields, they struggle to distinguish large-radius graphs like CFI graphs. Noting that the expressiveness of $S_3$ and $S_4$ is bounded by $N_1$ and $N_2$, respectively, as analyzed by Papp & Wattenhofer (2022). Conversely, $M_1$ is implemented by heterogeneous message passing, which makes it unaffected by large graph diameters, thus maintaining its performance across different graphs. SPD-WL is another 1-WL extension operated on a complete graph with shortest path distances as edge features. It may degrade to 1-WL on low-radius graphs, causing its relatively poor performance.

**Subgraph-based GNNs.** Regarding subgraph-based models, they can generally distinguish almost all Basic graphs, simple regular graphs and Extension graphs. However, an exception lies with NGNN, which performs poorly in Extension graphs due to its simplicial node selection policy and lack of node labeling. Two other exceptions are KP-GNN and I$^2$-GNN, both exhibiting exceptional performance in Regular graphs. KP-GNN can differentiate a substantial number of strongly regular

graphs and 4-vertex condition graphs, surpassing the 3-WL partially. And $I^2$-GNN surpasses the limitations of 3-WL partially through its enhanced cycle-counting power. An influential aspect that impacts the performance is the subgraph radius. Approaches incorporating appropriate encoding functions are expected to yield superior performance as the subgraph radius increases. However, in practice, enlarging the radius may result in the smoothness of information, wherein the receptive field expands, encompassing some irrelevant or noisy information. Hence, we treat the subgraph radius as a hyperparameter, fine-tuning it for each model, and present the best results in Table 2. Please refer to Appendix G for further details regarding the radius selection.

When comparing various subgraph GNNs, KP-GNN can discriminate part of the strongly regular graphs by peripheral subgraphs. Additionally, distance encoding in DE+NGNN and $I^2$-GNN enables better discrimination among different hops within a given subgraph radius, particularly in larger subgraph radii. As for DS-GNN, DSS-GNN, GNN-AK, SUN and SSWL_P, they employ similar aggregation schemes with slight variations in their operations. These models exhibit comparable performance, with SSWL_P outperforming others, which aligns with expectations that SSWL_P achieves the most expressiveness with least components (Zhang et al., 2023a).

$k$-**WL hierarchy-based GNNs.** For the $k$-WL-hierarchy-based models, we adopt two implemented approaches: high-order simulation and local-WL simulation. PPGN serves as the representative work for the former, while $\delta$-$k$-LGNN and KCSet-GNN as the latter. PPGN aligns its performance with 3-WL across all graphs except for CFI graphs. For CFI graphs with large radii, more WL iterations (layers of GNNs) are required. However, employing many layers may lead to over-smoothing, resulting in a gap between theoretical expectations and actual performance. Nonetheless, PPGN still surpasses most GNNs in CFI graphs due to global $k$-WL's global receptive field. For $\delta$-$k$-LGNN, we set $k = 2$, while for KCSet-GNN, we set $k = 3, c = 2$ to simulate local 3-WL, adhering to the original configuration. By comparing the output results with relatively small diameters, we observed that local WL matches the performance of general $k$-WL. However, local WL exhibits lower performance for CFI graphs with larger radii due to insufficient receptive fields.

**Substructure-based GNNs** For substructure-based GNNs, we select GSN, which incorporates substructure isomorphism counting as features. The best result obtained for GSN-e is reported when setting $k = 4$. For further exploration of policy and size, please refer to Appendix I.

**Random GNNs** Random GNNs are unsuitable for GI problems since even identical graphs can yield different outcomes due to inherent randomness. However, the RPC can quantify fluctuations in the randomization process, thereby enabling testing for random GNNs. We test DropGNN and OSAN. For more details regarding the crucial factor of random samples, please refer to Appendix J.

**Transformer-based GNNs** For transformer-based GNNs, we select Graphormer, which is anticipated to possess a level of expressiveness with SPD-WL. The experimental results verify that.

## 6 CONCLUSION AND FUTURE WORK

This paper proposes a new dataset, BREC, for GNN expressiveness measurement. BREC addresses the limitations of previous datasets, including difficulty, granularity, and scale, by incorporating 400 pairs of diverse graphs in four categories. A new evaluation method is proposed for principled expressiveness evaluation. Finally, a thorough comparison of 23 baselines on BREC is conducted. The experiment highlights a gap between theoretical and practical expressiveness. Additionally, the algorithms implemented in practice for the first time offer valuable tools for future research.

Apart from the expressiveness comparison based on GI, there are various other metrics for GNN expressiveness evaluation, such as substructure counting, diameter counting, and biconnectivity checking. However, it's worth noting that these tests are often conducted on datasets not specifically designed for expressiveness (Huang et al., 2023; Zhao et al., 2022a; Chen et al., 2020), which can lead to biased results caused by spurious correlations. In other words, certain methods may struggle to identify a particular substructure, but they can capture another property that correlates with substructures, resulting in false high performance. This problem can be alleviated in BREC because of the difficulty. We reveal the data generation process of BREC in Appendix L, hoping that researchers can utilize them in more tasks. We also hope the test of practical expressiveness will aid researchers in exploring its effects on performance in real datasets and other domains.

## 7 REPRODUCIBILITY

The datasets and evaluation codes are released at https://github.com/brec-iclr2024/brec-iclr2024. The GitHub repository provides the complete dataset and test methods, as well as the first implementation of the traditional algorithm for further use by researchers. In addition to complete and customizable testing via the GitHub repository, users can install the Pypi package for easier and faster testing. We commit to maintaining this dataset for the long term.

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

## A    DETAILS ON REGULAR GRAPHS

In this section, we introduce the relationship between four types of regular graphs. The inclusion relations of them are shown in Figure 4, but their difficulty relations and inclusion relations are not consistent.

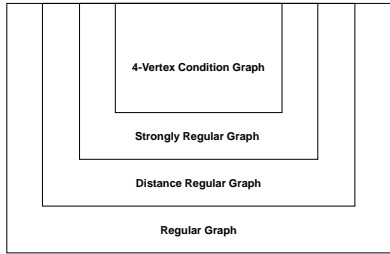

Figure 4: Regular graphs relationship

A graph is deemed a regular graph when all of its vertices possess an identical degree. If a regular graph, with $v$ vertices and degree $k$, satisfies the additional conditions wherein any two adjacent vertices share $\lambda$ common neighbors, and any two non-adjacent vertices share $\mu$ common neighbors, it is categorized as a strongly regular graph. Hence, it can be represented as $\mathrm{srg}(v, k, \lambda, \mu)$, denoting its four associated parameters.

Regular graphs and strongly regular graphs find wide application in expressiveness analysis. The difficulty of strongly regular graphs surpasses that of general regular graphs due to the imposition of additional requirements. Notably, the simplest strongly regular graphs with identical parameters $(\mathrm{srg}(16, 6, 2, 2))$ are exemplified by the Shrikhande graph and the $4 \times 4$-Rook's graph, as depicted in Figure 2(c).

Both 4-vertex condition graphs and distance regular graphs introduce heightened complexities, albeit in opposing directions. A 4-vertex condition graph is a strongly regular graph with an additional property that mandates the determination of the number of edges between the common neighbors of two vertices based on their connectivity. Conversely, distance regular graphs expand upon the definition of strongly regular graphs by specifying that for any two vertices $v$ and $w$, the count of vertices at a distance $j$ from $v$ and at a distance $k$ from $w$ relies solely on $j$, $k$, and the distance between $v$ and $w$. Notably, a distance regular graph with a radius of 2 is equivalent to a strongly regular graph.

The 4-vertex condition graph has yet to be explored in previous research endeavors. Similarly, instances of distance regular graphs are relatively scarce and analyzing them through examples proves to be challenging. To encourage further research in these domains, we have incorporated them into BREC.

## B    NODE FEATURES

In this section, we present the concept of node features and edge features in graphs.

We commence by providing the definition of graphs using an adjacency matrix representation. Consider a graph where the node features are represented by a $d_n$-dimensional vector, and the edge features are represented by a $d_e$-dimensional vector. This graph can be denoted as $\mathcal{G} = (\mathbf{V}(\mathcal{G}), \mathbf{E}(\mathcal{G}))$, where $\mathbf{V}(\mathcal{G}) \in \mathbb{R}^{n \times d_n}$ represents the node features, and $\mathbf{E}(\mathcal{G}) \in \mathbb{R}^{n \times n \times (d_e+1)}$ represents the edge

features, with $n$ being the number of nodes in the graph. The adjacency matrix of the graph is denoted as $\mathbf{A}(\mathcal{G}) \in \mathbb{R}^{n \times n} = \mathbf{E}(\mathcal{G})_{:,:,(d_e+1)}$, where $\mathbf{A}(\mathcal{G})_{i,j} = 1$ if $(i,j) \in \mathbb{E}(\mathcal{G})$ (i.e., if nodes $i$ and $j$ are connected by an edge), otherwise $\mathbf{A}(\mathcal{G})_{i,j} = 0$. The feature of node $i$ is represented by $\mathbf{V}(\mathcal{G})_{i,:}$, and the feature of edge $(i,j)$ is represented by $\mathbf{E}(\mathcal{G})_{i,j,1:d_e}$. The permutation (or reindexing) of $\mathcal{G}$ is denoted as $\mathcal{G}^\pi = (\mathbf{V}(\mathcal{G}), \mathbf{E}(\mathcal{G}))$ with permutation $\pi : [n] \to [n]$, such that $\mathbf{V}(\mathcal{G})_{i,:} = \mathbf{V}(\mathcal{G})_{\pi(i),:}$ and $\mathbf{E}(\mathcal{G})_{i,j,:} = \mathbf{E}(\mathcal{G})_{\pi(i),\pi(j),:}$.

Next, we explore the utilization of features. It is evident that incorporating node features during initialization and edge features during message passing can enhance the performance of GNNs, given appropriate hyperparameters and training. However, we should consider whether features can truly represent graph structures or provide additional expressiveness. Let us categorize features into two types.

The first type involves fully utilizing the original features, such as distances to other nodes or spectral embeddings. While using these features can aid GNNs in solving Graph Isomorphism (GI) problems, this type of feature requires a dedicated design to effectively utilize them. For instance, if we aim to recognize a 6-cycle in a graph, we can manually identify the cycle and assign distinct features to each node within the cycle. In this way, the GNN can recognize the cycle by aggregating the six distinctive features. However, the injecting strategy influences expressiveness and requires further analysis. Utilizing distance can also enhance expressiveness but also need a suitable design (like subgraph distance encoding and SPD-WL).

The second type entails incorporating additional features, such as manually selected node identifiers. it is important to note that this improvement stems from reduced difficulty rather than increased expressiveness. For instance, given a pair of non-isomorphic graphs with high similarity, we can manually find the components causing the distinguishing difficulty and assign identifiers to help models overcome them. However, this process is generally unavailable in practice.

In summary, we can conclude that features have the potential to introduce expressiveness, but this should be accomplished through model design rather than relying solely on the dataset. In the case of BREC, a dataset created specifically for testing expressiveness, we do not include additional meaningful features. Instead, we employ the same vector for all node features and edge features and adhere to specific model settings to incorporate graph-specific features, such as the distance between nodes in distance encoding based models.

## C    WL ALGORITHM

This section briefly introduces the WL algorithm and two high-order variants.

The 1-WL algorithm, short for "1-Weisfeiler-Lehman," is an initial version of the WL algorithm. It serves as a graph isomorphism algorithm and can be employed to generate a distinctive label for each graph.

In the 1-WL algorithm, every node in the graph maintains a state or color, which undergoes refinement during each iteration by incorporating information from the states of its neighboring nodes. As the algorithm progresses, the graph representation evolves into a multiset of node states, ultimately converging to a final representation.

To circumvent these examples, researchers have devised a technique to augment each node in the 1-WL test, resulting in the development of the $k$-WL test (Babai & Kucera, 1979; ?). The $k$-dimensional Weisfeiler-Lehman test expands the scope of the test to consider colorings of k-tuples of nodes instead of individual nodes. This extension allows for a more comprehensive analysis of graph structures and assists in overcoming the limitations posed by certain examples.

In addition to the $k$-WL test, Cai et al. (1989) proposed an alternative WL test algorithm that also extends to $k$-tuples. This variant is commonly referred to as the $k$-FWL ($k$-folklore-WL) test. The $k$-FWL test differs from the $k$-WL test in terms of how neighbors are defined and the order in which aggregation is performed on tuples and multisets.

There are three notable results associated with these tests:

     1  1-WL = 2-WL

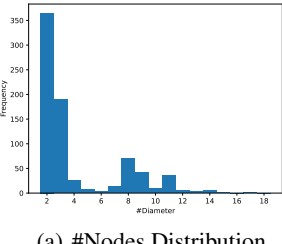
(a) #Nodes Distribution

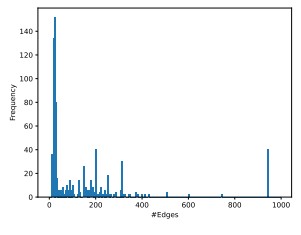
(b) #Edges Distribution

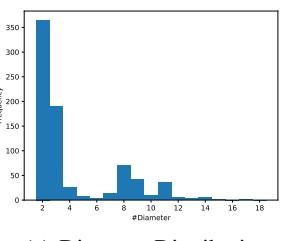
(c) Diameter Distribution

Figure 5: BREC Statistics

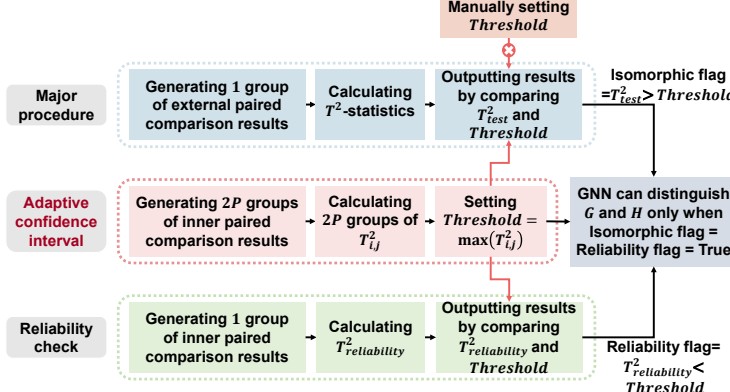

Figure 6: RAPC pipeline.

2  $k$-WL ¿ $(k-1)$-WL, $(k > 2)$

3  $(k-1)$-FWL = $k$-WL

More details can be found in Sato (2020); Huang & Villar (2021).

## D   CIRCULANT SKIP LINKS (CSL) GRAPHS

A CSL graph is defined as follows: Let $r$ and $m$ be co-prime natural numbers with $r < m - 1$. $\mathcal{G}(m, r) = (\mathbb{V}, \mathbb{E})$ is an undirected 4-regular graph with $\mathbb{V} = [m]$, where the edges form a cycle and include skip links. Specifically, for the cycle, $(j, j+1) \in \mathbb{E}$ for $j \in [m-1]$, and $(m, 1) \in \mathbb{E}$. For the skip links, the sequence is recursively defined as $s_1 = 1$, $s_{i+1} = (s_i + r) \bmod m + 1$, and $(s_i, s_{i+1}) \in \mathbb{E}$ for any $i \in \mathbb{N}$. Two Sample graphs with $m = 10, r = 2 \ or \ 3$ are shown in Fig 1(b).

In the CSL dataset, 10 CSL graphs with $m = 41$ and $r = 2, 3, 4, 5, 6, 9, 11, 12, 13, 16$ are generated. Thus resulting in 10 non-isomorphic but 1-WL-indistinguishable graphs.

## E   BREC STATISTICS

Here we give some statistics of the BREC dataset, shown in Figure 5.

## F   RAPC: A RELIABLE AND ADAPTIVE EVALUATION METHOD

In this section, we propose RAPC with an additional stage called adaptive confidence interval based on RPC. Though RPC performs excellently in experiments with a general theoretical guarantee in reliability, with manually setting $\alpha$. We still want to make the procedure more automated. In addition, we found that the inner fluctuations of each pair, i.e. $T^2_{\text{reliability}}$, vary from pairs. This means some graph outputs are more stable than others, and their threshold can be larger than others. However, it is impossible to manually set the confidence interval ($\alpha$) for all pairs, thus, we propose

an adaptive confidence interval method to solve this problem. The key idea is to set the threshold according to minimum internal fluctuations.

Given a pair of non-isomorphic graphs $\mathcal{G}$ and $\mathcal{H}$ to be tested. For simplicity, we rename $\mathcal{G}$ as $\mathcal{G}_1$, $\mathcal{H}$ as $\mathcal{G}_2$. For each graph ($\mathcal{G}_1$ and $\mathcal{G}_2$), we generate $p$ groups of graphs, with each group containing $2q$ graphs, represented by:

$$\mathcal{G}_{i,j,k}, \ i \in [2], \ j \in [p], \ k \in [2q]. \tag{10}$$

Similarly, we can calculate $T^2$-statistics for each group ($2p$ groups in total):

$$T_{i,j}^2 = q\overline{\boldsymbol{d}}_{i,j}^T \boldsymbol{S}_{i,j} \overline{\boldsymbol{d}}_{i,j}, \ i \in [2], \ j \in [p]. \tag{11}$$

where

$$\overline{\boldsymbol{d}}_{i,j} = \frac{1}{q}\sum_{k=1}^{q} \boldsymbol{d}_{i,j,k}, \ \boldsymbol{d}_{i,j,k} = f(\mathcal{G}_{i,j,k}) - f(\mathcal{G}_{i,j,k+q}), \ i \in [2], \ j \in [p], \ k \in [q],$$
$$\boldsymbol{S}_{i,j} = \frac{1}{q-1}\sum_{j=1}^{q}(\boldsymbol{d}_{i,j,k} - \overline{\boldsymbol{d}}_{i,j})(\boldsymbol{d}_{i,j,k} - \overline{\boldsymbol{d}}_{i,j})^T. \tag{12}$$

Similar to major procedure, we can conduct an $\alpha$-level test of $H_0 : \delta = \boldsymbol{0}$ versus $H_1 : \delta \neq \boldsymbol{0}$, it should always accept $H_0$(the GNN cannot distinguish them) since the $2q$ graphs in each group are essentially the same. And $T^2$-statistics should satisfy the:

$$T_{i,j}^2 = q\overline{\boldsymbol{d}}_{i,j}^T \boldsymbol{S}_{i,j} \overline{\boldsymbol{d}}_{i,j} < \frac{(q-1)n}{(q-n)} F_{n,q-n}(\alpha). \tag{13}$$

If the GNN can distinguish the pair, $T_{\text{test}}^2$ in major procedure and $T_{i,j}^2$ in adaptive confidence interval should satisfy the:

$$T_{\text{test}}^2 > \frac{(q-1)n}{(q-n)} F_{n,q-n}(\alpha) > T_{i,j}^2, \forall i \in [2], \ j \in [p]. \tag{14}$$

Thus we set the adaptive confidence interval as Threshold $= \text{Max}_{i\in\{1,2\}, \ p\in\{1,...,P\}}\{T_{i,p}^2\}$. Then we conduct Major Procedure and Reliability Check based on Threshold similar to RPC. The pipeline is shown in Fig 6.

In our analysis of the current evaluation method, we take into account the probabilities of false positives and false negatives. Typically, achieving extremely low levels of both probabilities simultaneously is challenging, and there is often a trade-off between them. However, since false positives can undermine the reliability of the methods, we prioritize establishing stringent bounds for this type of error. On the other hand, false negatives are explained in a more intuitive manner, acknowledging their presence but placing greater emphasis on minimizing false positives.

Regarding false positives, we give the following theorem.

**Theorem F.1** *The false positive rate with adaptive confidence interval is $\frac{1}{2^{2P}}$.*

**Proof F.1** *We first define false positives more formally. False positives mean the GNN $f$ cannot distinguish $\mathcal{G}$ and $\mathcal{H}$, but we reject $H_0$ and accept $H_1$. $f$ cannot distinguish $\mathcal{G}$ and $\mathcal{H}$ means $f(\mathcal{G}) = f(\mathcal{H}) = f(\mathcal{G}^\pi) \sim \mathcal{N}(\boldsymbol{\mu}_\mathcal{G}, \boldsymbol{\Sigma}_\mathcal{G})$. Since $\boldsymbol{d}_i$ in major procedure and $\boldsymbol{d}_{i,j,k}$ in adaptive confidence interval are derived from paired comparison by same function outputs, i.e., from $f(\mathcal{G})$ and $f(\mathcal{H})$, and from $f(\mathcal{G})$ and $f(\mathcal{G}^\pi)$, respectively. $\boldsymbol{d}_i$ and $\boldsymbol{d}_{i,j,k}$ should follow the same distribution, leading that $T_{test}^2$ and $T_{i,j}^2$ are independently random variables following the same distribution. Thus $P(T_{test}^2 > T_{i,j}^2) = \frac{1}{2}$. Then we can calculate the probability of false positives as*

$$P(Rejecting \ H_0) = P(T_{test}^2 > Threshold = Max_{i\in[2], \ j\in[p]}\{T_{i,j}^2\}) = \frac{1}{2^{2p}}. \tag{15}$$

*Thus we proof theorem F.1.*

Regarding false negatives, we propose the following explanation. A small threshold can decrease the false negative rate. Thus without compromising the rest of the theoretical analysis, we give the minimum value of the threshold. Equation 13 introduces a minimum threshold restriction. We obtain the threshold strictly based on it by taking the maximum value, which is the theoretical minimum threshold that minimizes the false negative rate.

Table 3: A general theoretical expressiveness upper bound of subgraph with radius $k$

| Radius | 1 | 2 | 3 | 4 | 5 | 6 | 7 | 8 | 9 | 10 |
|---|---|---|---|---|---|---|---|---|---|---|
| #Accurate on BREC | 252 | 298 | 300 | 327 | 326 | 385 | 398 | 398 | 399 | 400 |

Table 4: The performance of 3-WL with different iteration times

| Iterations | 1 | 2 | 3 | 4 | 5 |
|---|---|---|---|---|---|
| #Accurate on BREC | 193 | 209 | 217 | 264 | 270 |

## G  SUBGRAPH GNNS

In this section, we discuss settings for subgraph GNN models. The most important setting is the subgraph radius. As discussed before, a larger radius can capture more structural information, increasing the model's expressiveness. However, it will include more invalid information, making reaching the theoretical upper bound harder. Thus we need to find a balance between the two.

To achieve this, we first explore the maximum structural information that can be obtained under a given radius. Following Papp & Wattenhofer (2022), we implement $N_k$ method, which embeds the isomorphic type of $k$-hop subgraph when initializing. This method is only available in the theoretical analysis as one can not solve the GI problem by manually giving graph isomorphic type. We mainly use it as a general expressiveness upper bound of subgraph GNNs. The performance of $N_k$ on BREC is shown in Table 3. Actually, $N_3$ already successfully distinguishes all graphs except for CFI graphs. $k = 6$ is an important threshold as $N_k$ outperforms 3-WL (expressiveness upper bound for most subgraph GNNs (Frasca et al., 2022; Zhang et al., 2023a)) in all types of graphs. An interesting discovery is that increasing the radius does not always lead to expressiveness increasing as expected. This is caused by the fact that we only encode the exact $k$-hop subgraph instead of 1 to $k$-hop subgraphs. This phenomenon is similar to subgraph GNNs, revealing the advantages of using distance encoding.

We then test the subgraph GNNs' radii by increasing them until reaching the best performance, which is expected to be a perfect balance. For some methods, radius= 6 is the best selection, which is consistent with the theory. The exceptions are NGNN, NGNN+DE, KPGNN, I²-GNN and SSWL_P. NGNN directly uses an inner GNN to calculate subgraph representation, whose expressiveness is restricted by the inner GNN. As the subgraph radius increases, though the subgraph contains information, the simple inner GNN can hardly give a correct representation. That's why radius= 1 is the best setting for NGNN. NGNN+DE and I²-GNN add distance encodings, making the subgraph with a large radius can always clearly extract a subgraph with a small radius. Therefore, a large radius= 8 is available. KPGNN utilizes a similar setting by incorporating distance to subgraph representation, and radius= 8 is also the best setting. KPGNN can also use graph diffusion to replace the shortest path distance. Though graph diffusion outperforms some graphs, the shortest path distance is generally a better solution. Previous findings reveal the advantages of using distance, which we hope can be more widely used in further research. SSWL_P achieves better expressiveness with theoretical minimum components, making more information available.

## H  $k$-WL HIERARCHY GNNS

In this section, we discuss settings for $k$-WL hierarchy GNN models. $k$-WL algorithm requires a converged tuple embedding distribution for GI. However, $k$-WL hierarchy GNNs do not have the definition of converging. It will output the final embeddings after a specific number of layers, i.e., the iteration times of $k$-WL. Thus we need to give a suitable number of layers where the $k$-WL converged after the number of iteration times. In theory, increasing the number of layers always leads to a non-decreasing expressiveness, since the converged distribution will not change furthermore. However, more layers may cause over-smoothing, leading to worse performance in practice.

Table 5: Substructure-based model performance on BREC

| Model | Basic Graphs (60) | | Regular Graphs (140) | | Extension Graphs (100) | | CFI Graphs (100) | | Total (400) | |
|---|---|---|---|---|---|---|---|---|---|---|
| | Number | Accuracy | Number | Accuracy | Number | Accuracy | Number | Accuracy | Number | Accuracy |
| $S_3$ | 52 | 86.7% | 48 | 34.3% | 5 | 5% | 0 | 0% | 105 | 26.2% |
| $S_4$ | 60 | 100% | 99 | 70.7% | 84 | 84% | 0 | 0% | 243 | 60.8% |
| GSN-v(k=3) | 52 | 86.7% | 48 | 34.3% | 5 | 5% | 0 | 0% | 105 | 26.2% |
| GSN-v(k=4) | 60 | 100% | 99 | 70.7% | 84 | 84% | 0 | 0% | 243 | 60.8% |
| GSN-e(k=3) | 59 | 98.3% | 48 | 34.3% | 52 | 52% | 0 | 0% | 159 | 39.8% |
| GSN | 60 | 100% | 99 | 70.7% | 95 | 95% | 0 | 0% | 254 | 63.5% |

Table 6: The performance of DropGNN with different sample numbers

| #Samples | 100 | 200 | 400 | 800 | 1200 | 1600 |
|---|---|---|---|---|---|---|
| #Accurate on BREC | 177 | 222 | 242 | 253 | 260 | OOM |

To keep a balance, we utilize similar methods for subgraph GNNs. We first analyze the iteration times of 3-WL, shown in Table 4. One can see 6 iteration times are enough for all types of graphs. Then we increase the layers of $k$-WL GNNs until reaching the best performance. We finally set 5 layers for PPGN, 4 layers for KCSet-GNN and 6 layers for $\delta$-k-LGNN.

## I  SUBSTRUCTURE-BASED GNNS

In this section, we discuss the performance of substructure-based GNN models. Specifically, we focus on the GSN (Graph Substructure Network) model proposed by Bouritsas et al. (2022), which offers a straightforward neural network implementation, denoted as GSN-v, of the $S_k$ substructure. Additionally, we introduce GSN-e, a slightly stronger version of GSN-v that incorporates features on edges instead of just nodes.

Experimental results presented in Table 5 demonstrate that GSN-v achieves a perfect match with the performance of $S_k$. Furthermore, GSN-e outperforms GSN-v, indicating superior performance when edge features are included.

## J  RANDOM GNNS

In this section, we delve into the settings for random GNNs. Random GNNs leverage samples from graphs using specific strategies, and both the number of samples and the sampling strategies have an impact on performance.

For DropGNN, the sampling strategy revolves around a relatively straightforward approach of deleting nodes. As for the number of samples, it is recommended to set it to the average number of nodes in the dataset. In our reported results, we set the number of samples to 100, which aligns with the average number of nodes. The ablation study results on the number of samples can be found in Table 6.

Another approach, OSAN, proposes a data-driven method that achieves similar performance with fewer samples. This is achieved by training the model to select diverse samples. However, it requires an additional training framework and may not necessarily lead to improved performance. In our case, we select the edge-deleting strategy and set the number of samples to 20.

## K  EXPERIMENT SETTINGS

All experiments were performed on a machine equipped with an Intel Core i9-10980XE CPU, an NVIDIA RTX4090 graphics card, and 256GB of RAM.

Table 7: Model Hyperparameters

| Model | Radius | Layers | Inner dim | Learning rate | Weight decay | Batch size | Epoch | Early stop threshold |
|---|---|---|---|---|---|---|---|---|
| NGNN | 1 | 6 | 16 | $1e-4$ | $1e-5$ | 32 | 20 | 0.01 |
| DE+NGNN | 8 | 6 | 128 | $1e-4$ | $1e-5$ | 32 | 30 | 0.01 |
| DS-GNN | 6 | 10 | 32 | $1e-4$ | $1e-5$ | 32 | 30 | 0 |
| DSS-GNN | 6 | 9 | 32 | $1e-4$ | $1e-4$ | 32 | 20 | 0.01 |
| SUN | 6 | 9 | 32 | $1e-4$ | $1e-4$ | 32 | 20 | 0.01 |
| SSWL_P | 8 | 8 | 64 | $1e-5$ | $1e-5$ | 8 | 20 | 0.1 |
| GNN-AK | 6 | 4 | 32 | $1e-4$ | $1e-4$ | 32 | 10 | 0.1 |
| KP-GNN | 8 | 8 | 32 | $1e-4$ | $1e-4$ | 32 | 20 | 0.3 |
| I²GNN | 8 | 5 | 32 | $1e-5$ | $1e-4$ | 16 | 20 | 0.2 |
| PPGN | / | 5 | 32 | $1e-4$ | $1e-4$ | 32 | 20 | 0.2 |
| $\delta$-k-LGNN | / | 6 | 16 | $1e-4$ | $1e-4$ | 16 | 20 | 0.2 |
| KC-SetGNN | / | 4 | 64 | $1e-4$ | $1e-4$ | 16 | 15 | 0.3 |
| GSN | / | 4 | 64 | $1e-4$ | $1e-5$ | 16 | 20 | 0.1 |
| DropGNN | / | 10 | 16 | $1e-3$ | $1e-5$ | 16 | 100 | 0 |
| OSAN | / | 8 | 64 | $1e-3$ | $1e-5$ | 16 | 40 | 0 |
| Graphormer | / | 12 | 80 | $2e-5$ | 0 | 16 | 100 | 0 |

**RPC settings.** For non-GNN methods, the output results are uniquely determined, and as such, this part of the experiment does not require RPC. It is worth noting that most non-GNN baselines involve running graph isomorphism testing software on subgraphs, and they mainly serve as theoretical references in our evaluation.

Regarding GNNs, we employ RPC with $q = 32$ and $d = 16$ to evaluate their performance. Considering a confidence level of $\alpha = 0.95$, which is a typical setting in statistics, the threshold should be set to $\frac{(q-1)d}{(q-d)} F_{d,q-d}(\alpha) = 31 F_{16,16}(0.95) = 72.34$.

To ensure robustness, we repeat all evaluation methods ten times using different seeds selected from the set $\{100, 200, \ldots, 1000\}$. We consider the final results reliable only if the model passes the Reliability check for all graphs with any seed, meaning that the quantification of the output embedding distance between isomorphic pairs is always smaller than the threshold. The reported results are selected as the best results rather than the average, as we aim to explore the upper bound of expressiveness.

**Training settings.** We employ a Siamese network design and utilize the cosine similarity loss function. Another commonly used loss function is contrastive loss (Hadsell et al., 2006), which directly calculates the difference between two outputs. However, we opt for cosine similarity loss due to its advantage of measuring output difference under the same scale through normalization. This approach prevents model outputs from being excessively amplified, which could otherwise magnify minor precision errors and treat them as differentiated results of the model.

We use the Adam optimizer with a learning rate searched from $\{1e-3, 1e-4, 1e-5\}$, weight decay selected from $\{1e-3, 1e-4, 1e-5\}$, and batch size chosen from $\{8, 16, 32\}$. Graphormer, on the other hand, follows the original training settings on ZINC.

We incorporate an early stopping strategy, which halts training when the loss reaches a small value. While for random GNNs, we do not utilize early stopping. The maximum number of epochs is typically set to around 20 since the model can often distinguish a pair relatively quickly.

**Model hyperparameters.** The most crucial hyperparameters related to expressiveness, such as the subgraph radius for subgraph GNNs and the number of layers for $k$-WL hierarchy GNNs, are determined through theoretical analysis, as outlined in Appendix G and H. These hyperparameters have a direct impact on the expressiveness of the models.

Other hyperparameters also implicitly influence expressiveness. We generally adopt the same settings as previous expressiveness datasets, with two exceptions: inner embedding dimension and batch normalization.

The inner embedding dimension reflects the model's capacity. For smaller and simpler expressiveness datasets used in the past, a small embedding dimension has been sufficient. However, the

appropriate embedding dimension for BREC is unknown, so we generally conduct a search within the range of $16, 32, 64, 128$.

Additionally, we utilize batch normalization for all models, even though it may not have been used in all previous models. Batch normalization helps control the outputs within a suitable range, which can be beneficial for distinguishing graph pairs.

The detailed hyperparameter settings for each method are provided in Table 7.

## L  GRAPH GENERATION

In this section, we provide an overview of how the graphs in the BREC dataset were generated.

**Basic graphs.** This category consists of 60 pairs of graphs, each containing 10 nodes. To generate these graphs, the 1-WL algorithm was applied to all 11.7 million graphs with 10 nodes, resulting in a hash value for each graph. Among these graphs, 83,074 happened to have identical hash values as others. From this set, 60 pairs of graphs were randomly selected.

**Regular graphs.**  This category includes 140 pairs of regular graphs.  For the 50 simple regular graphs, the search was conducted for regular graphs with 6 to 10 nodes, and 50 pairs of regular graphs with the same parameters were randomly selected.  For the 50 strongly regular graphs, the number of nodes ranged from 16 to 35.  The graphs were obtained from sources such as http://www.maths.gla.ac.uk/ es/srgraphs.php and http://users.cecs.anu.edu.au/ bdm/data/graphs.html. For the 20 4-vertex condition graphs, a search was conducted on http://math.ihringer.org/srgs.php, and the simplest 20 pairs of 4-vertex condition graphs with the same parameters were selected. For the 20 distance regular graphs, a search was performed on https://www.distanceregular.org/, and the simplest 20 pairs of distance regular graphs with the same parameters were chosen.

**Extension graphs.** This category consists of 100 pairs of graphs based on comparing results between GNN extensions. The $S_3$, $S_4$, and $N_1$ algorithms were applied to all 1-WL-indistinguishable graphs with 10 nodes. This yielded 4,612 $S_3$-indistinguishable graphs, 1,132 $N_1$-indistinguishable graphs, and 136 $S_4$-indistinguishable graphs. From these sets, 60 pairs of $S_3$-indistinguishable graphs, 20 pairs of $N_1$-indistinguishable graphs, and 10 pairs of $S_4$-indistinguishable graphs were randomly selected. Care was taken to ensure that no graphs were repeated. Additionally, 10 pairs of graphs were added using a virtual node strategy, including 5 pairs obtained by adding a virtual node to a 10-node regular graph and 5 pairs based on $C_{2l}$ and $C_{l,l}$ as described in Papp & Wattenhofer (2022).

**CFI graphs.** This category consists of 100 pairs of graphs generated based on the CFI methods proposed by Cai et al. (1989). All CFI graphs with backbones ranging from 3 to 7-node graphs were generated. From this set, 60 pairs of 1-WL-indistinguishable graphs, 20 pairs of 3-WL-indistinguishable graphs, and 20 pairs of 4-WL-indistinguishable graphs were randomly selected.

These different categories of graphs provide a diverse range of graph structures and properties for evaluating the expressiveness of GNN models.

