# OpenReview forum: "Towards Better Evaluation of GNN Expressiveness with BREC Dataset"
_ICLR.cc/2024/Conference — Submitted to ICLR 2024_

### Official Review · Reviewer_D7WE · 2023-10-24

**Soundness:** 3 good
**Presentation:** 3 good
**Contribution:** 1 poor
**Rating:** 5
**Confidence:** 4

**Summary:**

The paper tackles the problem of benchmarking GNNs, especially being concerned with their expressive power. The authors propose the BREC dataset to tackle this problem. The dataset aims to fix some issues in existing datasets, in particular, the authors argue that existing benchmarks are not very granular and too easy with models either achieving perfect accuracy or random guessing. The authors further propose an evaluation technique that is more principled than existing techniques, that takes also into account numerical errors.

**Strengths:**

The authors are very thorough in their work and consider a number of interesting points when designing the dataset. I particularly enjoyed the section that takes into account numerical errors induced for instance by floating point arithmetic errors. I think that the authors propose a principled approach to tackle this issue that could be valuable in general, outside of the scope of this specific work as well.

**Weaknesses:**

While the work is valuable, I believe that it is not novel enough in its current state. The tasks are very synthetic and even though I agree with the authors that the BREC dataset seems more interesting than existing benchmarks such as CSL, in my opinion it seems to be a marginal improvement over such datasets in terms of the novelty factor. There is definitely a need for such datasets in the community and the new evaluation technique is interesting, but I am not convinced that in its current state the work is fit for ICLR. Overall, this feels more of an "extension" to current datasets than something very novel.

**Questions:**

Would the authors be able to clarify any interesting findings that come from Table 2? It is a bit challenging for me to spot any significant trends the way it is currently ordered. It might be useful to further group the models based on their type.

Would it be possible to clarify the contributions of the work? I understand that the work is proposing a new dataset and a new evaluation technique for it (which I find to be interesting and valid), but are there further novel contributions?

---

> ### Author Response · Authors · 2023-11-18
> **Response to Reviewer D7WE Part 1/2**
>
> We thank reviewer D7WE for acknowledging the intricate design and insightful evaluation techniques. We answer your questions and address your concerns as follows:
>
> > W1: While the work is valuable, I believe that it is not novel enough in its current state. The tasks are very synthetic and even though I agree with the authors that the BREC dataset seems more interesting than existing benchmarks such as CSL, in my opinion it seems to be a marginal improvement over such datasets in terms of the novelty factor. There is definitely a need for such datasets in the community and the new evaluation technique is interesting, but I am not convinced that in its current state the work is fit for ICLR. Overall, this feels more of an "extension" to current datasets than something very novel.
>
> R1:
>
> We thank the reviewer for highlighting a pivotal criterion concerning benchmarks. BREC is not limited to an upgraded version of previous datasets but provides many additional perspectives in expressive power analysis. A critical difference between BREC and previous datasets is that BREC is a benchmark for testing "practical expressiveness" for the first time. However, previous datasets only serve to verify "theoretical expressiveness". We explain their difference in the following.
>
> 1. Previous expressiveness analysis mainly focused on "theoretical expressiveness", which represents the theoretical upper bound of the model under strong assumptions (deep enough layers, universal approximation of MLP, etc.). However, whether these models' practical implementations can achieve their theoretical power is constantly questioned and worth studying. Previous expressiveness datasets only served as a simple and incomplete verification due to their toy and repetitive components, making models tend to easily achieve 100% accuracy. Still, BREC is designed to test the "practical expressiveness", i.e., whether the theoretically powerful models can practically reach their expressiveness under more diverse, more difficult and larger scale settings than previous datasets.
> 2. The "practical expressiveness" results also lead to many interesting findings. For detailed analysis, please refer to R2.
>
> Further, we would like to highlight again that BREC fundamentally differs from existing datasets in multiple aspects:
>
> 1. New evaluation method. Previous datasets do not consider evaluation techniques from an expressiveness perspective. They assign a label for each graph and consider it a classification task. It fundamentally restricts extending a toy dataset to a practical one.
> 2. Diversity in graphs. Previous toy datasets have too few components/graphs, where one set of hyperparameters can achieve perfect accuracy. BREC has many more pairs of distinct non-isomorphic graphs advanced in difficulty, granularity, and scale, which can test models' true expressivity.
> 3. Worthwhile experiment results. BREC not only verifies theoretical expressiveness but also provides more insightful findings. For example, BREC can test models without theoretical expressiveness characterizations. Half of the baselines we tested in Table 2 do not have a precise expressiveness analysis, but our experiment results make the quantification and comparison feasible. In contrast, previous datasets are restricted to 1-WL and 3-WL bounded models as their graphs are designed strictly following 1-WL/3-WL distinguishability. They cannot compare models between 1-WL and 3-WL, or beyond 3-WL.

---

> ### Author Response · Authors · 2023-11-18
> **Response to Reviewer D7WE Part 2/2**
>
> >Q1: “Would the authors be able to clarify any interesting findings that come from Table 2? It is a bit challenging for me to spot any significant trends the way it is currently ordered. It might be useful to further group the models based on their type.”
>
> R2:
>
> We thank the reviewer for the valuable suggestion. We have revised Table 2 in the updated paper by grouping each type of model. We also summarize the findings below.
>
> 1. There is gap between theoretical expressiveness and practical accuracy in many models. For example, PPGN is theoretically the same as 3-WL. However, our tests firstly show that it cannot reach its theoretical expressive power, even weaker than some models proven to be less powerful than 3-WL (e.g., SSWL_P). The gap can also explain why some models with high theoretical expressiveness do not perform well in real-world graphs.
> 2. Some techniques are very useful for improving the practical expressiveness. For example, distance encoding [1] greatly improves the performance of NGNN+DE, KPGNN, I2-GNN, SSWL_P, etc. In comparison, node labeling, which is proved to bring the same expressiveness improvement for subgraph-based models [2] as distance encoding, do not perform as well in practice. These findings can inspire new research on designing practically more powerful positional/structural encodings.
> 3. Generally, higher expressive power leads to better real-world performance. For example, subgraph-based models show excellent expressiveness results in our experiments, which align well with their state-of-art real-world performance [3]. Our results can serve as an efficient tool for analysis. A more detailed analysis can be found at R2 to reviewer ZHhj.
>
> References
>
> [1] Distance Encoding: Design Provably More Powerful Neural Networks for Graph Representation Learning https://arxiv.org/abs/2009.00142
>
> [2] A Complete Expressiveness Hierarchy for Subgraph GNNs via Subgraph Weisfeiler-Lehman Tests http://arxiv.org/abs/2302.07090
>
> [3] Towards Arbitrarily Expressive GNNs in O(n2) Space by Rethinking Folklore Weisfeiler-Lehman https://arxiv.org/abs/2306.03266
>
> >Q2: “Would it be possible to clarify the contributions of the work? I understand that the work is proposing a new dataset and a new evaluation technique for it (which I find to be interesting and valid), but are there further novel contributions?”
>
> R3:
>
> We thank the reviewer for the valuable question. Other than the new dataset and evaluation technique (our major contributions), we summarize some other contributions as follows.
>
> 1. Discovering flaws of existing datasets: Previous datasets have limitations in their difficulty, granularity and scale. They cannot be used to evaluate newly proposed expressive models. As we pointed out in motivation, we are the first to discover inherent flaws of current datasets and propose a solution.
> 2. Revealing the gap between "theoretical expressiveness" and "practical expressiveness": Our dataset provides many additional perspectives in expressive power analysis. By revealing the gap between theoretical and practical expressiveness, we can find the influence of different strategies, components, frameworks, and other factors that may be underestimated.
> 3. We give the first and most thorough empirical expressiveness measurement results. The quantification is especially useful for models without precise expressiveness analysis. By comparing it with broader experiment results on real-world datasets, we can also capture what is needed regarding expressiveness. We hope users can utilize it and aid further research.
>
> Finally, we would like to respectfully point out that ICLR welcomes a broad range of subject areas, including datasets and benchmarks (explicitly listed in topics). As a benchmark paper, our BREC has a clear motivation, intricate designs, and a sound evaluation pipeline, and it also addresses an urgent need of the community, which makes us believe that BREC meets the acceptance bar of ICLR. We appreciate the review's invaluable questions and advice, and kindly request the reviewer to reevaluate the contribution and novelty of our paper, possibly from a benchmark paper standard. We are profoundly thankful for your time and effort!

---

> > ### Comment · Reviewer_D7WE · 2023-11-20
> >
> > I thank the authors for the thorough answer. I have thought about this for some time. I also thank the authors for grouping models in Table 2, I think it is a significant improvement.
> >
> > I agree with the points raised by the authors, but in my opinion, the improvement remains marginal over what exists in the literature. I also agree with the authors that the proposed evaluation procedure is interesting, but it feels more like a "nice-to-have" rather than tackling a concrete existing issue in current evaluations on expressiveness benchmarks.
> >
> > The fundamental issue I see is that generating such datasets is not very challenging in regard to the effort necessary to collect such data or label it. There is of course non-trivial effort involved in your work to do such a job in a diligent manner, but I'm not convinced that it's a sufficiently novel benchmark, when compared to what currently exists.
> >
> > Regardless, I have raised my score as I do believe that the improved Table and the discussion have improved the work. I finally thank the authors for their effort in their rebuttal.

---

> > > ### Author Response · Authors · 2023-11-21
> > > **Acknowledging Your Feedback: Offering a Deeper Look into Our Work (Part 1/2)**
> > >
> > > We sincerely appreciate your thoughtful comments and the time you invested in evaluating our work. Your feedback is invaluable, and we are grateful for the positive aspects you highlighted, including the improved Table 2 and the thoroughness of our response. We also acknowledge your concerns about the perceived challenges and effort in generating our expressiveness dataset, especially compared to the effort of human labeling in standard benchmark papers. We would like to provide further insight into the extensive effort involved in creating the graphs in our dataset, which is highly challenging and nontrivial.
> > >
> > > Concretely, for Extension Graphs, we implemented four distinct extension methods within our framework: the k-WL hierarchy, k-hop subgraph, counting substructure, and node marking. These methods were uniformly integrated into the WL framework, and we systematically applied them in iterative tests on graphs that met specific criteria. Additionally, we introduced a sub-sampling approach to lower testing burden and balance different types of graphs. To further enhance difficulty, we incorporated dedicated examples featuring the use of virtual nodes and large cycles. In the case of CFI Graphs, we pioneered the implementation of CFI Graphs generation and rigorously tested them using k-WL algorithms. Our work on regular graphs also extended beyond conventional benchmarks by not only a wider search for rare regular graphs (please refer to Appendix L) but also by introducing new sub-categories like the 4-vertex condition graph. Additionally, we implemented theoretical upperbound algorithms for certain models, such as the distance-WL algorithm for Graphormer, and incorporated various graph representation techniques, including incidence graphs and hamming codes, because some rare regular graphs do not provide graph compositions but generation methods with other representations.
> > >
> > > We have made our implemented algorithms publicly available at https://github.com/brec-iclr2024/brec-iclr2024/tree/main/Non-GNNs and https://github.com/brec-iclr2024/brec-iclr2024/tree/main/dataset_generation. We would like to especially highlight our contribution to implementing CFI graphs. As pointed out by reviewer Bpfn, "The authors are the first to implement CFI graphs that enable any k-WL tests." Our implementation enables generating CFI graphs meeting **arbitrary k-WL test difficulty** and using **arbitrary backbone graphs**. This is not only superficially beneficial for GNN expressiveness comparison right now, but also provides a profound contribution to the broader research community, including the graph theory community, as CFI graphs are an essential category for studying the graph isomorphism problem but have never been implemented before due to their implementation challenges. CFI graphs are crafted initially with backbone graphs, necessitating the determination of an appropriate separator size. Subsequently, the original nodes, along with their combinations, undergo expansion, resulting in a multitude of nodes. The creation of new edges is facilitated by leveraging information from the original edges and incorporating cardinal parity considerations. The final two non-isomorphic graphs are generated with additional twist operations. These operations maintain an underlying similarity while introducing nuanced differences between the graphs, which make them distinguishable by some k+1-WL test but not by k-WL. Furthermore, the complexity of the generated graphs is rigorously validated through the application of high-order WL algorithms, which we also implemented ourselves for arbitrary k for the first time. In addition, the involved operations in the implementation are generally different from standard graph operators, and thus are missing from existing graph libraries, which forced us to implement these graph operators one by one ourselves too (see https://github.com/brec-iclr2024/brec-iclr2024/blob/main/dataset_generation/utils.py for the util functions we implemented). All these challenges demonstrate the nontrivial effort in generating our dataset. Overall, the CFI graphs quickly grow very large, and we automate this process for arbitrary k and arbitrary backbone graphs, which can emancipate researchers from composing them by hand. This involves not only nontrivial intellectual effort but also facilitates future research on graph isomorphism testing, graph theory, and GNN expressivity.

---

> > > ### Author Response · Authors · 2023-11-21
> > > **Acknowledging Your Feedback: Offering a Deeper Look into Our Work (Part 2/2)**
> > >
> > > We hope this detailed explanation underscores the tremendous effort we have spent in the dataset generation, as well as the technical challenges we have overcome and the valuable contributions that distinguish our work from other expressiveness benchmarks. Finally, we would like to highlight the difference between human-labeled datasets and synthesized datasets (such as our BREC). Although the former one takes great effort in manual labeling, the data samples are usually easy to collect. In contrast, the latter one (synthesized dataset) takes great effort in generating the data samples while usually does not require labeling. The two kinds of datasets have different aspects of difficulty and challenges, but are both nontrivial to collect.
> > >
> > > Once again, we extend our gratitude for your constructive feedback, and we appreciate that you recognized the improvements in our work. We look forward to any further discussions.

---

> > > ### Author Response · Authors · 2023-11-23
> > > **Grateful for Reviewers’ comments and a Gentle Reminder**
> > >
> > > We hope this letter finds you well. Firstly, we want to express our sincere gratitude for your time and expertise in reviewing our work. Your insightful feedback has been invaluable in improving the quality of our research. We highly appreciate your attention to detail and thoughtful evaluation.
> > >
> > > We would like to kindly remind you about our response to your previous comments and concerns. We have carefully considered and addressed them in our response and corresponding code repository. Specifically, we have dedicated a discussion to elaborate on our efforts in generating the dataset, emphasizing the methodologies employed and providing clear access to public codes for further analysis and usage.
> > >
> > > Once again, we want to extend our heartfelt thanks for your time and insightful feedback. Your contributions have truly made a difference in our study. We eagerly await your comments. Should you require any additional information or have any further questions, please do not hesitate to reach out to us.
> > >
> > > Thank you once again for your invaluable support.

---

### Official Review · Reviewer_Bpfn · 2023-10-31

**Soundness:** 3 good
**Presentation:** 3 good
**Contribution:** 3 good
**Rating:** 6
**Confidence:** 3

**Summary:**

The paper overcomes the limitations of previous expressiveness datasets in terms of difficulty, granularity, and scale by introducing 4 datasets. Each dataset covers different benchmarking purposes for comprehensive GNN expressiveness evaluations. The authors further introduce Reliable Paired Comparisons instead of applying traditional classification comparisons to eliminate possible spurious correlations that can lead to unfair comparisons.

**Strengths:**

1. The paper is the first benchmark that can cover different difficulties with fine granularity.
2. The authors are the first to implement CFI graphs that enable any k-WL tests.
3. Applying pair-wise comparisons instead of classifications to eliminate influence from other factors is reasonable and rigorous.
4. To overcome the dilemma between false negative and false positive, the authors propose RPC that includes similarity comparisons and internal fluctuation considerations. Moreover, the authors propose to adjust the similarity threshold adaptively.
5. The code is well-organized.

**Weaknesses:**

The overall benchmark is comprehensive and elaborated. I only have a few minor concerns/questions.
1. Why do the authors adopt cosine similarity instead of other contrastive loss? Will this loss be possible to introduce any spurious correlation leading to biased results?
2. Since PPGN was compared, I'm wondering why "INVARIANT AND EQUIVARIANT GRAPH NETWORKS" was not included in the comparisons.

**Questions:**

N/A

---

> ### Author Response · Authors · 2023-11-18
> **Response to Reviewer Bpfn**
>
> We appreciate Reviewer Bpfn's valuable questions and address the concerns as follows:
>
> >W1: Why do the authors adopt cosine similarity instead of other contrastive loss? Will this loss be possible to introduce any spurious correlation leading to biased results?
>
> R1:
>
> We appreciate the reviewer's meticulous examination of our evaluation methodology.Our choice of cosine similarity in the contrastive learning setting is notably simplified compared to common scenarios where batches of samples need to be separated. In our case, the objective is to distinguish only one sample from another, eliminating the necessity for certain properties, particularly the control of differences in one type. Unlike typical contrastive learning, which aims to maintain alignment and uniformity, and often introduces additional terms to regulate the distance between samples in one batch, such considerations are rendered unnecessary in our context. Consequently, we adhere to the simplest method to streamline the evaluation process.
>
> >W2: Since PPGN was compared, I'm wondering why "INVARIANT AND EQUIVARIANT GRAPH NETWORKS" was not included in the comparisons.
>
> R2:
>
> We appreciate the reviewer's observation regarding the exclusion of "INVARIANT AND EQUIVARIANT GRAPH NETWORKS" from our comparisons. While this model introduces invariant and equivariant layers and holds significance in expressiveness analysis [1, 2, 3], the original model it proposes, IGN, does not exhibit high expressiveness. As highlighted in [1], the k-IGN variant can implement k-WL but is constrained by the need for iterating k-th and 2k-th bell numbers' operations (the 4th bell number is 15, and the 6th bell number is 203), limiting its applicability and scalability to k <= 2. Furthermore, 2-WL is only as expressive as 1-WL, indicating that 2-IGN cannot distinguish any pairs of graphs in BREC. Consequently, we have chosen not to include IGN in our testing. Instead, we have evaluated several models inspired by IGN, such as PPGN [1], ESAN (DSS-GNN) [2], SUN [3], among others.
>
> References
>
> [1] Provably Powerful Graph Networks https://arxiv.org/abs/1905.11136
>
> [2] Equivariant Subgraph Aggregation Networks https://arxiv.org/abs/2110.02910
>
> [3] Understanding and Extending Subgraph GNNs by Rethinking Their Symmetries https://arxiv.org/abs/2206.11140

---

> > ### Comment · Reviewer_Bpfn · 2023-11-20
> >
> > Thank you for your response. I lean to accept the paper but not confidently. The overall idea and contributions are interesting with the first implementation of CFI graphs that enable any k-WL tests. But as Reviewer D7WE said, this benchmark is "nice-to-have" but not necessary.

---

> > > ### Author Response · Authors · 2023-11-21
> > > **Gratitude for Your Feedback and a Slight Clarification**
> > >
> > > Thank you ever so much for your thoughtful acknowledgment of our paper. We are sincerely grateful for the time and effort you dedicated to reviewing our work.
> > >
> > > While addressing your comments, we noted a slight clarification that may be necessary regarding the characterization of our benchmark as "nice-to-have." We would like to emphasize that the "nice-to-have" sentiment primarily applies to using our new evaluation techniques on previous benchmarks. Specifically, these datasets are inherently simpler, and employing a classification task that assigns each graph an individual label proves sufficient for their evaluation.
> > >
> > > However, our evaluation technique holds paramount importance for our new benchmark with advancements in difficulty, granularity, scale, and essential data arrangement. It provides a more precise depiction of a model's expressiveness, crucial for in-depth expressiveness analysis in certain situations, as opposed to relying on vague and random accuracy. Moreover, our proposed benchmark is urgently needed for the research community, as previous benchmarks fall short of aligning with recent trends and evolving model requirements.
> > >
> > > Thank you once again for your insightful review. We hope this clarification aligns with the intended perspective. If you have any further questions or concerns, we would be more than happy to address them.

---

### Official Review · Reviewer_ZHhj · 2023-11-01

**Soundness:** 3 good
**Presentation:** 2 fair
**Contribution:** 2 fair
**Rating:** 6
**Confidence:** 2

**Summary:**

Previous datasets designed for evaluating the expressiveness of GNNs had limitations in terms of difficulty, granularity, and scale. The authors propose a new expressiveness dataset called BREC, which includes 400 pairs of non-isomorphic graphs carefully selected from different categories. The dataset offers higher difficulty, finer granularity, and larger scale compared to previous datasets. The authors conduct experiments on 23 models with expressiveness beyond 1-WL (Weisfeiler-Lehman) on the BREC dataset, providing a thorough measurement of their expressiveness and highlighting the gap between theoretical and practical expressiveness.

**Strengths:**

* It is a valuable problem for evaluating the expressiveness of GNNs.
* The benchmark is extensive, including datasets with higher difficulty, finer granularity, larger scale and many models for comparisons.
* It is promising for the BREC dataset to serve as a benchmark for testing the expressiveness of future GNNs.

**Weaknesses:**

1. One concern is about the evaluation. It shows in Table 2 that only 400 samples are included in total. Including more samples in the datasets would make the experiments more convincing.

2. What about the relationship about the performance on the proposed synthetic datasets and real-world datasets? In other words, is the model that  shows to more expressive in the benchmark performing better in real-world tasks? It would be better to provide more fine-grained analyses.

**Questions:**

See weaknesses.

---

> ### Author Response · Authors · 2023-11-18
> **Response to Reviewer ZHhj**
>
> We appreciate Reviewer ZHhj's insightful comments and address the concerns as follows:
>
> > W1: One concern is about the evaluation. It shows in Table 2 that only 400 samples are included in total. Including more samples in the datasets would make the experiments more convincing.
>
> R1:
>
> We thank the reviewer for emphasizing the importance of dataset size. We carefully considered how to construct the datasets initially and decided to use 400 pairs for coming reasons:
>
> 1. 400 samples are not small for expressiveness. Recalling that each sample includes 2 quite similar but non-isomorphic graphs, and all 800 graphs are non-isomorphic. Thus, generating 319,600 pairs of graphs (from CSL and SR25 settings) or even more by adding noisy components (from EXP settings) will be easy. However, we limit the number to 400 pairs (each graph appears only once) to keep the most essential and difficult comparisons.
> 2. Unlike large-scale real-world datasets, which require as many graphs as possible splited into train/val/test for testing generalization and scalability, our datasets aim to test expressiveness, where we do not need the model to generalize by training on big data. Instead, expressiveness datasets purely aim to measure models’ distinguishing power. Furthermore, we provide a reliable and precise evaluation method to obtain the exact distinguishable number of graphs with nearly **no variance**. This indicates that the current number of graphs is **sufficient** to reflect the expressiveness gap between different methods.
> 3. While it is feasible to expand the datasets by seamlessly sampling more graphs, certain graph types, such as Basic and Extension, lend themselves to exhaustive search sampling. However, this approach is not universally applicable, particularly for graphs like CFI or distance regular graphs, which are too rare to procure in larger quantities. Furthermore, enlarging the dataset in this manner may lead to an unbalanced distribution, disproportionately favoring simple graphs. Consequently, we have chosen to retain a set of 800 graphs.
>
> > W2: What about the relationship about the performance on the proposed synthetic datasets and real-world datasets? In other words, is the model that shows to more expressive in the benchmark performing better in real-world tasks? It would be better to provide more fine-grained analyses.
>
> R2:
>
> We thank the reviewer for underscoring the significance of practical utility concerning expressiveness. Generally, graphs in chemical areas tend to have high expressiveness demand. There exist some structures that standard MPNNs can hardly recognize. For example, the benzene ring (6-cycle) is a common and essential molecule component that most simple GNNs (including all MPNNs) cannot recognize. It is inevitable those GNNs might learn some statistical patterns correlated to benzene rings, but they fundamentally fail to understand the structure and fail to make predictions causally. In real-world datasets, expressive GNNs also perform remarkably in molecular datasets. ZINC [1] is a widely used molecular property regression benchmark. Currently, the 2 SOTA models are GRIT [2] and N$^2$-GNN [3], both focusing on expressiveness.
>
> We collect our tested models' performance on ZINC:
>
> | Model | BREC(all) | BREC(CFI Graph) | ZINC↓ |
> | --- | --- | --- | --- |
> | OSAN | 148 | 5 | 0.155 |
> | Graphormer | 79 | 10 | 0.122 |
> | DS-GNN | 222 | 16 | 0.116 |
> | GSN | 254 | 0 | 0.115 |
> | DE+NGNN | 231 | 21 | 0.111 |
> | DSS-GNN | 221 | 15 | 0.097 |
> | GNN-AK | 222 | 15 | 0.093 |
> | KP-GNN | 275 | 11 | 0.093 |
> | SUN | 223 | 13 | 0.083 |
> | I$^2$-GNN | 281 | 21 | 0.083 |
> | PPGN | 233 | 23 | 0.079 |
> | KC-SetGNN | 211 | 1 | 0.075 |
> | SSWL_P | 248 | 38 | 0.070 |
> | N$^2$-GNN | 287 | 27 | 0.059 |
>
>
> Observably, N$^2$-GNN and SSWL_P outperform other models on the ZINC dataset. Both models exhibit commendable performance on BREC, particularly on CFI Graph, known for its substantial size and asymmetric composition, necessitating the model to discover graph pattern differences in its layers iteratively. The ZINC dataset, with its regression target "constrained solubility," incorporating water-octanol partition coefficient, synthetic accessibility score, and the number of cycles with more than six atoms, requires structure learning from a molecular accessibility perspective. Expressiveness, therefore, plays a pivotal role in enhancing model performance in the ZINC dataset.
>
> References
>
> [1] Benchmarking Graph Neural Networks https://arxiv.org/abs/2003.00982
>
> [2] Graph Inductive Biases in Transformers without Message Passing http://arxiv.org/abs/2305.17589
>
> [3] Towards Arbitrarily Expressive GNNs in O(n2) Space by Rethinking Folklore Weisfeiler-Lehman https://arxiv.org/abs/2306.03266

---

### Author Response · Authors · 2023-11-20
**Looking Forward to Your Reply**

Dear Reviewers,

We sincerely appreciate your time and effort in reviewing our manuscript! Your insightful questions and valuable feedback have been extremely helpful. We have endeavored to address these points thoroughly in our detailed rebuttal.

As the discussion period is drawing to a close in the next three days, we kindly request that you review our responses at your earliest convenience. Your further comments and guidance will be highly beneficial to the refinement of our work.

Thank you very much for your continued attention to our paper. We eagerly await your valuable feedback.

Best regards,
Authors

---

### Meta-Review · Area_Chair_DEL5 · 2023-12-19

**Metareview:**

This paper proposed BREC, a new expressiveness dataset to benchmark GNN Expressiveness. Reviewers found this paper to be interesting and useful, but have concerns related to novelty and the significance of the contribution.

**Justification For Why Not Higher Score:**

The novelty concern from reviewers seems to be valid

**Justification For Why Not Lower Score:**

N/A

---

### Decision · Program_Chairs · 2024-01-16

Reject